# Heterogeneity of Cancer Stem Cells in Tumorigenesis, Metastasis, and Resistance to Antineoplastic Treatment of Head and Neck Tumours

**DOI:** 10.3390/cells10113068

**Published:** 2021-11-08

**Authors:** Nicola Cirillo, Carmen Wu, Stephen S. Prime

**Affiliations:** 1Melbourne Dental School, The University of Melbourne, Carlton, VIC 3053, Australia; wuc1@student.unimelb.edu.au; 2Centre for Immunology and Regenerative Medicine, Institute of Dentistry, Barts and the London School of Medicine and Dentistry, Queen Mary University of London, London E1 4NS, UK; stephensprime@gmail.com

**Keywords:** cancer stem cells, head and neck tumours, CD44, ALDH, CD133

## Abstract

The discovery of a small subset of cancer cells with self-renewal properties that can give rise to phenotypically diverse tumour populations has shifted our understanding of cancer biology. Targeting cancer stem cells (CSCs) is becoming a promising therapeutic strategy in various malignancies, including head and neck squamous cell carcinoma (HNSCC). Diverse sub-populations of head and neck cancer stem cells (HNCSCs) have been identified previously using CSC specific markers, the most common being CD44, Aldehyde Dehydrogenase 1 (ALDH1), and CD133, or by side population assays. Interestingly, distinct HNCSC subsets play different roles in the generation and progression of tumours. This article aims to review the evidence for a role of specific CSCs in HNSCC tumorigenesis, invasion, and metastasis, together with resistance to treatment.

## 1. Introduction

Head and neck squamous cell carcinoma (HNSCC) is a broad term that encompasses a variety of malignancies arising in the oral cavity, nasal cavity, paranasal sinuses, pharynx, and larynx [1]. HNSCC is a growing global burden and is the sixth most common type of cancer worldwide [1]. These cancers carry a poor prognosis, with a 5-year survival rate for all stages combined being approximately 50–60% [2]. The mortality of HNSCC is mainly caused by local recurrence and cervical lymph node metastasis and occasionally, by distant organ metastasis as well as by treatment failure or acquisition of resistance to therapy [3].

Cancer research in the 20th century has been dominated by the stochastic model of carcinogenesis, which proposes that all the cells in the tumour population have an equal potential in acquiring and accumulating genetic and/or epigenetic mutations in critical genes that control cell growth and differentiation. This results in the generation and selection of clone(s) with proliferative advantages, which eventually acquire the ability to invade surrounding tissues [4,5].

However, there is increasing support for the concept that not all cells within a tumour are equal and only a subpopulation of cells have the ability to self-renew and generate new tumours—the cancer stem cells (CSCs) [4,5]. The CSC hypothesis postulates that cells within a tumour are organised within a hierarchy, at the apex of which is a small subset of cells that possess the stem cell qualities of self-renewal and can give rise to phenotypically diverse tumour populations via asymmetrical differentiation [4]. A CSC, therefore, is a cell within a tumour that possesses the capacity to self-renew and to give rise to the heterogeneous lineages of cancer cells that comprise the original tumour [3]; hence, CSCs are tumorigenic or tumour forming. Interestingly, not all CSCs are equal, and distinct subpopulations exist that can lead to functionally different processes.

In HNSCC, the existence of CSCs has been well reported in the literature [6]. Head and neck cancer stem cells (HNCSCs) have been previously identified using CSC-specific markers, the most common being CD44, Aldehyde Dehydrogenase 1 (ALDH1), and CD133 [3]. The side population (SP) assay is another way of identifying CSCs; the SP phenotype is determined by the ability to efflux a dye through an ATP-binding cassette (ABC) membrane transporter [7]. SP cells have been identified in HNSCCs and shown to have stem-like properties and it is now known that different stemness signatures correlate with distinctive molecular and functional properties of CSCs [7,8,9,10], thus highlighting the importance of CSC heterogeneity in carcinogenesis.

As our knowledge of CSCs continues to grow, the therapeutic targeting of CSCs in cancer patients is being evaluated. This article aims to review the role of CSC heterogeneity—as defined by the expression of a specific phenotype—in tumorigenesis, metastasis, and the radio-, chemo-, and immune resistance of HNSCC.

## 2. Cancer Stem Cells’ Heterogeneity in Head and Neck Cancer

A recent scoping review identified CSC-associated molecules that are commonly used as CSC markers [3]. In cancers of the head and neck, these markers included CD44, ALDH1, CD133, Oct3/4, Nanog, and SOX2; a single common CSC sorting marker could not be found. This supports the notion that distinct subpopulations of CSCs might be involved in different pathophysiological functions during the multistep process of carcinogenesis, although phenotypic heterogeneity in terms of marker expression does not necessarily entail different functions. The canonical functions of commonly used CSC-associated markers are reported in Appendix A.

The role of CSCs in malignancy has been investigated in vitro, in vivo, and in clinical studies. In particular, the expression of specific CSC markers and/or stemness genes has been correlated with histological grade, stage, and the presence of metastasis in HNSCC patients. The models of CSC function vary widely and include two-dimensional human and animal cell lines/strains, three-dimensional organoids, and the use of immunodeficient mice. In this Section, we have summarised the evidence to support a role for CSCs in tumorigenesis, metastasis, and radio-, chemo-, and immune-resistance (Table 1, Table 2 and Table 3). The type of study, the study population, and the experimental findings of each study supporting the association between a stemness phenotype and carcinogenesis are listed in Appendix A.

### 2.1. Tumorigenesis

Compelling evidence supports a role of CSCs in HNSCC tumorigenesis (Table 1). CSCs have been identified via a single or a combination of stem cell markers, including CD44, ALDH1, CD133, c-Met, or via the SP assay. Only CSCs have an increased tumorigenic potential following xenograft transplantation to mice and when the number of transplanted cells are examined, a much smaller number of CSCs are able to initiate tumour formation compared to non-CSCs [7,8,9,10,11,12,13,14,15,16,17,18,19,20,21,22,23,24,25,26]. Taken together, the data indicate that HNSCCs contain a small subpopulation of cells that: (1) have an enhanced ability for self-renewal, as demonstrated using colony and sphere formation assays (Appendix A) [7,8,9,10,11,12,13,17,18,19,20,21,22,23,26,27,28,29,30,31,32,33], (2) are capable of generating tumours that mimic the heterogeneity of the parental tumours, and (3) maintain tumorigenicity following multiple in vivo passages [9,13,14,17,21,23,28,34].

**Table 1 cells-10-03068-t001:** **Enhanced tumorigenic properties of Head and Neck Cancer Stem Cells.** Experimental findings supporting the enhanced self-renewal, proliferative, and tumorigenic potential of head and neck cancer stem cells.

Ref.	Authors(s), Year	Type of Study	Cancer Stem Cell Subpopulations
[11]	Chen et al., 2009	in vitro and SCID mouse model	ALDH1+, CD44+CD24−, CD44+CD24-ALDH1+
[34]	Adams et al., 2015	in vitro and SCID mouse model	ALDH activity and CD44+ (ALDHhighCD44high)
[26]	Lim et al., 2012	in vitro and BALB/c mouse model	CSC-like phenotype
[15]	Chinn et al., 2015	in vitro and NOD/SCID mouse; Cross sectional study	CD44high, CD44high/ALDH1+, CD44low/ALDH-
[16]	Lee et al., 2017	in vitro and in vivo (B10; B6-Rag2−/−II2rg−/− mouse models)	CD44+, CD44+PD-L1-, CD44+PD-L1+
[19]	Wu et al., 2017	in vitro and NOD/SCID mouse model	CD44+, CD44+/CD133+
[27]	Seino et al., 2016	in vitro	CD44+, CD44high/ALDH1high or ALDH1low
[9]	Song et al., 2010	in vitro and nude mouse model	SP * and non-SP cells
[22]	Lim et al., 2014	in vitro and in vivo mouse model	ALDHhigh, ALDHlow
[28]	Krishnamurthy et al., 2010	in vitro and in vivo SCID mouse model	ALDH+CD44+Lin−, ALDH−CD44−Lin−
[29]	Campos et al., 2012	in vitro	ALDH+CD44+, ALDH−CD44−
[30]	Chen et al., 2011	in vitro	ALDH1+, ALDH1-
[31]	Okamoto et al., 2009	in vitro	CD44+, CD133+CD44+, CD44+CD133+ABCG-2+
[21]	Sun and Wang, 2011	in vitro and in vivo NOD/SCID mouse model	c-Met+, c-Met+/CD44+, c-Met-/CD44-
[20]	Bourguignon et al., 2012	in vitro	CD44v3highALDH1high CD44v3lowALDH11ow
[32]	Han et al., 2014	in vitro and in vivo nude mouse model	CD24+/CD44+, CD24-/CD44+
[18]	Prince et al., 2007	in vitro and in vivo NOD/SCID and Rag2DKO mouse model	CD44+, CD44-, CD44Lin-
[23]	Kaseb et al., 2016	in vitro and in vivo (Rag-2/γc−/−) mouse model	CSC (via spheroid forming assay), CD44+/CD66−, CD44+/CD66+
[25]	Wei et al., 2009	in vitro and in vivo SCID mouse model	CD133+, CD133-
[12]	Chen et al., 2010	in vivo (SCID mice and/or nude mice (BALB/c strain) mouse model	ALDH1+, ALDH1-
[7]	Yanamoto et al., 2011	in vitro	SP cell fraction
[24]	Chiou et al., 2008	in vitro and in vivo BALB/c nude mouse model; Cross-sectional study	Oral Cancer-Stem Like Cells (OC-SLC)
[8]	Zhang et al., 2009	in vitro and in vivo BALB/C nude mouse model	SP cells
[14]	Clay et al., 2010	in vitro and in vivo NOD/SCID mouse model	ALDHhigh, CD44+
[10]	Yu et al., 2016	in vitro and in vivo BALB/c nude mouse model	OSCC *-derived side population (OSCC-SP), OSCC-MP
[13]	Todaro et al., 2010	in vitro and in vivo combined immunocompromised mouse	ALDHhigh, ALDHlow

* OSCC = oral squamous cell carcinoma, * SP = side population.

### 2.2. Metastasis

The evidence for a role of CSCs in metastasis has been reported in association with the enhanced invasion potential of CSCs compared to non-CSCs in HNSCCs, both in vitro [9,11,12,31,32] and in vivo [15,24,35]; interestingly, it has also been reported that CSCs have a decreased ability to invade as assessed via the Matrigel Invasion Assay in vitro [36]. The CSC signatures used in these studies are shown in Table 2.

In brief, HNCSCs identified via markers (CD44, ALDH1, c-Met, SOX2) or SP assays have been shown to have enhanced metastatic potential in vivo [9,13,15,19,21,36]. Several studies [15,19,35,37,38,39] have also reported an enhanced expression of the stem cell markers CD44, ALDH1, c-Met, Oct3/4 and SOX-2 in primary tumours of patients with lymph node metastasis compared to those with negative lymph nodes (Table 2), although the differences were not always statistically significant [15]. Highly metastatic cell lines were also found to have a higher proportion of SP cells compared to low metastatic cell lines [9] (Appendix A).

**Table 2 cells-10-03068-t002:** **Enhanced invasive and metastatic properties of Head and Neck Cancer Stem Cells.** Experimental findings supporting a role head and neck cancer stem cells in enhancing invasive and metastatic potential.

Ref.	Author(s), Year	Type of Study	Cancer Stem Cell Subpopulations
[15]	Chinn et al., 2015	in vitro and in vivo NOD/SCID mouse model; Cross-sectional study	CD44high, CD44high/ALDH1, CD44low, CD44low/ALDH-
[19]	Wu et al., 2017	in vitro and in vivo NOD/SCID mouse model	CD44+, CD44−
[9]	Song et al., 2010	in vitro and in vivo nude mouse model	SP, non-SP
[30]	Chen et al., 2011	in vitro	Spheroid Derived Cells (SDC)
[31]	Okamoto et al., 2009	in vitro	CD44+, CD44-
[21]	Sun and Wang 2011	in vitro and in vivo NOD/SCID mouse model	c-Met+
[37]	Michifuri et al., 2012	in vitro and cross-sectional study	ALDH1 expression, SOX2 staining
[32]	Han et al., 2014	in vitro and in vivo nude mouse model	CD24+/CD44+, CD24-/CD44+
[36]	Davis et al., 2010	in vitro and in vivo NOD/SCID mouse model	CD44high, CD44low
[35]	Lim et al., 2012	in vitro and in vivo BALB/c nude mouse model; Cross-sectional study	c-Met expression
[24]	Chiou et al., 2008	in vitro and in vivo BALB/c nude mouse model; Cross-sectional study	Oral Cancer-Stem Like Cells (OC-SLC)
[13]	Todaro et al., 2010	in vitro and in vivo immunocompromised mouse	ALDHhigh, ALDH1+

### 2.3. Radio-, Chemo, and Immune Resistance

Evidence for a role of CSC subpopulations in radio- and chemo-resistance has been reported in HNSCC (Table 3). HNCSCs have been shown to have enhanced survival after irradiation treatment [11,12,23,32] and also, to have a level of drug resistance to chemotherapeutic agents such as cisplatin, bortezomib, etoposide, 5-FU, and doxorubicin [7,9,21,31,32]. In one example, Han et al. (2014) identified two subpopulations of CSCs expressing CD44, namely, CD24-/CD44+ and CD24+/CD44+ cells, with the latter cell population having more chemo-resistance to gemcitabine and cisplatin compared to CD24-/CD44+ cells [32].

An increased expression of certain stemness markers such as Oct4, Sox2, and Nanog has been demonstrated in radio- and chemo-resistant HNSCC cells [40,41,42] and, finally, the CSC-like signature CD44 has been reported to be significantly correlated with an incomplete response to radio- and chemotherapy in patients with locally advanced HNSCC [43]. Evidence for a role of CSCs in immune resistance in HNSCC also exists. In particular, it has been reported that HNCSCs contribute to the downregulation of antitumor immunity in the tumour microenvironment [16].

**Table 3 cells-10-03068-t003:** **Enhanced radio-chemo and immune resistance of Head and Neck Cancer Stem Cells.** Experimental findings supporting the enhance radio-chemo resistance and upregulation of drug resistant and anti-apoptotic genes, as well as genes involved in immune regulation and in head and neck cancer stem cells.

Ref.	Author	Type of Study	Experimental Findings
[11]	Chen et al., 2009	in vitro and SCID mouse model	ALDH1+, CD44+CD24-ALDH1+, ALDH1-, CD44+CD24-ALDH1-
[43]	Koukourakis et al., 2012	in vitro, Cross-sectional study	CD44+
[9]	Song et al., 2010	in vitro and in vivo nude mouse model	SP, non-SP, ABCG2 expression
[31]	Okamoto et al., 2009	in vitro	ABCG1/2 expression and CD44+/CD44-
[21]	Sun and Wang 2011	in vitro and in vivo NOD/SCID mouse model	c-Met1
[20]	Bourguignon et al., 2012	in vitro	CD44v3highALDH1high
[32]	Han et al., 2014	in vitro and in vivo nude mouse model	CD24+/CD44+, CD24-/CD44+
[23]	Kaseb et al., 2016	in vitro and in vivo (Rag-2/γc−/−) mouse model	CSC (spheroids)
[12]	Chen et al., 2010	in vivo (SCID mice and/or nude mice (BALB/c) mouse models	ALDH1+
[7]	Yanamoto et al., 2011	in vitro	ABCG2 expression, SP, non-SP
[40]	Tsai et al., 2011	in vitro	CD133, CD117, ABCG2 expression; Nanog, Oct4, Bmi1, and Nestin
[41]	Chikamatsu et al., 2012	in vitro	CD44+, CD44-
[42]	Park et al., 2016	in vitro	Sox2, Nanog, CSCs derived from spheroids
[8]	Zhang et al., 2009	in vitro and in vivo BALB/C nude mouse model	SP, non-SP, ABCG1/2 expression
[10]	Yu et al., 2016	in vitro and in vivo BALB/c nude mouse model	CD133, SP, ABCG2, ALDH1
[16] *	Lee et al., 2017	in vitro and in vivo B10 and B6-Rag2−/− II2rg−/− mouse models	CD44+
[51] *	Chikamatsu et al., 2011	in vitro	CD44+, CD44-

Articles pertaining immune resistance are marked with an asterisk.

## 3. Molecular Features of HNCSCs Related to Tumorigenesis, Metastasis, and Resistance

In Section 2, we highlighted the research supporting an association between a distinct CSC phenotype, particularly the expression of certain CSC-like markers, and specific HNSCC characteristics, e.g., the expression of stemness markers Oct4, Sox2, and Nanog in chemo-resistant HNSCC cells. In Section 3, we review the putative mechanisms by which the molecules and pathways expressed and activated in CSC-like populations drive tumorigenesis, metastasis, and resistance to treatment in HNSCC. Whilst an extensive array of molecules and pathways involved in these functions have been published, we focus here on recurrent themes [44,45,46,47,48,49,50,51,52,53].

### 3.1. Upregulation of Genes Related to Self-Renewal

A number of studies have reported that stemness genes (Oct-4, Nanog, Sox2, Klf4, Bmi-1, and Nestin) are upregulated in CSCs isolated from cell lines and clinical HNSCCs [7,11,12,20,22,24,26,27,30,38,39,40,41,42]. These proteins may play important roles in maintaining the self-renewal capacity of CSCs which has implications for tumour growth and differentiation (Appendix A).

#### 3.1.1. Oct4-Sox2-Nanog Axis

Oct4, also known as Oct3 or Oct3/4, is encoded by *POU5F*, which belongs to the POU family of transcription factors; it is a nuclear protein that regulates the self-renewal and differentiation of embryonic stem cells. Oct4 expression in adult stem cells with an undifferentiated phenotype helps maintain the pluripotency of the cell population [44,45]. The SOX (sex-determining region Y [SRY]-box) family of transcription factors are well-established regulators of cell fate during development and are crucial for the derivation of embryonic stem cells (ESCs) from the inner cell mass and for the maintenance of ESCs [44]. Nanog is a homeodomain transcription factor that helps maintain the self-renewal capacity of ESCs; it plays a critical role in regulating the fate of the inner cell mass during embryonic development and functionally blocks differentiation [24,45].

Bourguignon and colleagues in 2012 demonstrated a strong correlation between the overexpression of CD44v3-Oct4-Sox2-Nanog and HNSCC progression [20]. CD44 is a receptor for hyaluronan (HA), a major component in the extracellular matrix in most mammalian tissues. Hence, CD44 provides a physical link between the extracellular matrix and transcription factors such as Oct4, Sox2, and Nanog that, in turn, regulate tumour cell function (20). Bourguignon et al. (2012) demonstrated that the interaction between HA and CD44 promoted tumour sphere formation, self-renewal, and clonal formation in CD44v3high/ALDH1high cells and also tumour formation in vivo, possibly via the overexpression of miR-302a and miR-302b [20]. The miR axis functions to downregulate lysine-specific histone demethylases (AOF1 and AOF2) and DNA (cytosine-5)-methyltransferase 1 (DNMT1) to induce global DNA demethylation [20].

Tsai et al. (2014) found that Oct4 overexpression enhanced cell proliferation, invasiveness, and colony formation in vitro and tumorigenicity and mesenchymal traits in vivo [39]. Lee et al. (2014) demonstrated that the overexpression of Sox2 in ALDH1 high cells enhanced tumour-sphere formation, self-renewal properties, and invasive potential via modulation of an epithelial to mesenchymal transition (EMT) [44]. In a consistent way, Sox2 downregulation reduces invasive potential through a mesenchymal to epithelial transition (MET), enhances chemosensitivity and inhibits xenograft tumour growth in vivo [44].

In clinical and prognostic studies, the expression of Oct4 and Nanog have been shown to be significantly correlated with a higher incidence of OSCC and lymph node metastasis [43,45]. Similarly, Michifuri et al. (2012) reported that a Sox2 diffuse staining pattern of tumour cells was significantly correlated with the presence of lymph node metastasis and the histological grade of OSCC [37]. Chiou et al. (2008) also showed that Oct4 and Nanog were correlated with the tumour stage but not with lymph node metastasis in OSCC [24]. Taken together, the data suggest that these proteins play a key role in the invasion and progression of OSCC and, possibly, are related to the metastatic potential of HNCSCs.

#### 3.1.2. Klf4, Bmi-1, and Nestin

Bmi-1 is a member of the Polycomb (PcG) family of transcriptional repressors that mediate gene silencing by regulating chromatin structure; it is critical for the maintenance and self-renewal in both normal stem cells and CSCs [12]. Bmi-1 expression has been shown to be enhanced in HNCSCs in multiple studies [8,11,18,21,28,39,41]. Bmi-1 knockdown suppresses the expression of stemness genes, including Oct-4, Nanog, Sox2, Nestin, as well as c-Myc and β-catenin in HNSCC ALDH1+ cells, and leads to a reduction in foci formation, migration, invasion, and tumour sphere formation [8,11,18,21,28,39,41], together with a downregulation of the drug-resistant genes MDR-1, MRP-1, and ABCG2 [12] (Appendix A). Taken together, the data support a role for Bmi-1 in the maintenance of self-renewal, tumorigenicity and chemo- and radio-resistance of HNCSCs.

#### 3.1.3. c-Met

c-Met is a transmembrane receptor tyrosine kinase that binds hepatocyte growth factor (HGF). HGF is predominantly secreted by mesenchymal cells and the binding of HGF to c-Met results in the activation of multiple signalling pathways related to cell proliferation, invasion and apoptosis resistance [22]. Lim et al. (2014) showed that the knockdown of c-Met attenuated CSC traits in HNSCC and augmented cisplatin chemosensitivity by suppressing the expression of the ABCG2 transporter gene. Further, subcutaneous and orthotopic xenograft tumour formation was inhibited, suggesting that the activation of c-Met is critical for the proliferation and maintenance of CSC traits in HNSCC [22]. In a previous study, Lim et al. (2012) had shown that the overexpression of c-Met promoted tumour growth and enhanced invasion and metastasis in vivo [35]. The data indicate that c-Met signalling plays an important role in tumour progression, invasion and metastasis.

#### 3.1.4. GSK3β

The enzyme glycogen synthase kinase 3β (GSK3β) is involved in the regulation of cell cycle progression and cell proliferation [33]. Under various pathological conditions, upstream signalling pathways such as PI3K/AKT, Raf/MEK/ERK, and Wnt induce the phosphorylation and inactivation of GSK3b [33]. GSK3b is upregulated in HNCSCs and is required to maintain phenotypes such as CD44high/ESAlow, CD44high/ESAhigh and CD44high/ALDH1high in HNCSC [27,33]. The inhibition of GSK3b reduces self-renewal potential, tumour sphere formation and the mRNA expression of the stem cell markers Sox2, Oct4, and Nanog leading to the induction of cell differentiation [27,33]. Further, Shigeishi et al. (2013) demonstrated that the knockdown of the normally high levels of active GSK3b in CD44high/ESAlow cells resulted in a more rapid shift of these cells into the CD44high/ESAhigh phenotype [33]. Signalling through GSK3b, therefore, may be a key regulator of the c-Met shift that enables metastatic tumour cells to produce new tumours at secondary sites. The preservation of functionally active GSK3b is required for CSC self-renewal and this is promoted by signalling pathways initiated by CD44 and RHAMM. A RHAMM (receptor for hyaluronic acid-mediated motility), such as CD44, binds HA. Both RHAMM and CD44 knockdowns result in the phosphorylation (inactivation) of GSK3b and the phosphorylation (activation) of ERK1/2, phenomena that may be related to a CD44-induced inhibition of the phosphorylation of AKT, thereby preventing AKT from phosphorylating and inactivating GSK3b [33].

Collectively, the data show that CSCs in HNSCC have upregulated the expression of genes and proteins related to maintaining stemness or self-renewal which, in turn, enhances the tumorigenic and metastatic phenotype of these cells.

### 3.2. Wnt/β-Catenin Signalling

Canonical Wnt signalling is a pathway engaged in the formation of head and neck tissues; it is not, however, active in adult somatic mucosal cells [52]. Importantly, canonical Wnt signalling regulates self-renewal mechanisms during normal stem cell development. β-catenin is a major component of the Wnt pathway, and, in the normal state, β-catenin is predominantly present as an Axin-APC-β-catenin complex that is phosphorylated by GSK3 β [46]. Following activation, β-catenin separates from the complex, enters into the nucleus and binds to transcription factors of T-cell factor. There follows upregulation of genes associated with proliferation and migration such as CyclinD1, c-myc, and Cox-2 [45,46].

Research shows that the Wnt/β-catenin pathway is activated in CSCs [9,26,45,46,47] (Appendix A). The mechanisms of Wnt/β-catenin pathway activation and dysregulation, together with the functional significance of Wnt/β-catenin signalling in HNSCC, have been reviewed recently [52]. Ravindran et al. (2015) demonstrated a significant correlation between β-catenin expression and both the tumour stage and lymph node metastasis in OSCC, findings that suggest a reactivation of stem cell pathways and a role for the dysregulated expression of these proteins in the progression and aggressiveness of OSCC [45].

Lim et al. (2012), Lee et al. (2014) and Warrier et al. (2014) all reported that the inhibition of the Wnt pathway reduced the stemness properties of HNCSCs, resulting in a reduced self-renewal capacity, spheroid formation, chemo-resistance and suppression of stemness markers including Oct4, Sox2, Nestin, CD44 and ALDH1 [26,46,47]. Lee et al. (2014) further demonstrated that the downregulation of β-catenin reduced the tumorigenic potential of CSCs in xenografted mice, whilst the overexpression of β-catenin promoted the proliferation of HNSCC cells and generated HNSCC cells with stem-like features [47]. This action was thought to occur partly through direct Oct4 regulation by β-catenin because Oct4 overexpression abrogated the inhibition of stemness induced by β-catenin knockdown [47].

Warrier et al. (2014) reported that Wnt/β-catenin inhibition also downregulated the transcription factors associated with EMT, namely Twist and Snail, and resulted in the accumulation of E-cadherin and reduction of N-cadherin with a reversal of EMT [46]. The loss of E-cadherin results in the release of β-catenin into the cytosol and elicits activation of the canonical Wnt signalling pathway [46]. Dysregulation of the Wnt pathway, therefore, can predispose cells to undergo EMT, leading to an increased potential for invasion and metastasis.

Together, these studies support a role for the canonical Wnt/β-catenin pathway in the self-renewal of CSCs, with the consequent impact on the tumorigenic, metastatic and treatment-resistance properties of HNCSCs.

### 3.3. Epithelial-Mesenchymal Transition

EMT is a developmental process in which epithelial cells acquire a migratory, mesenchymal phenotype, leading to the loss of cell polarity and cell–cell adhesion, the downregulation of epithelial markers (E-cadherin), and the upregulation of EMT-inducing transcription factors [17]. EMT is thought to constitute an important step in mediating the invasion and metastasis of epithelial cancers [17].

Numerous studies have shown that EMT-related genes and proteins (Twist, Snail, Bmi-1, Axl, Vimentin, N-cadherin, α-SMA) are differentially upregulated in CSCs [11,16,27,30,33] (Appendix A), whilst E-cadherin is downregulated [11,17,27,30,48]. The data indicate that CSCs reflect a “plastic state” of tumour cells. The capacity to undergo EMT is triggered by a variety of cellular and/or environmental signals.

Biddle et al. (2011) isolated two distinct populations of CSCs in OSCC that were either preferentially migratory or proliferative and, by examining the expression of the epithelial specific antigen ESA or EpCAM in a CD44+ cell populations, they identified two subsets of cells that were either ESAlow or ESAhigh [17]. CD44high/ESAlow cells were isolated from both cell lines and fresh OSCC tumour samples and were localised in situ at the periphery of colonies, exhibited enhanced sphere formation, and had a greater expression of EMT markers (Vimentin, Twist, Snail, Axl) and a lower expression of epithelial-specific genes (E-cadherin, Calgranulin B, Involucrin, Keratin 15). This cell type, therefore, appears to have an EMT phenotype and with a migratory phenotype and metastatic potential, it has the capacity to infiltrate lymph nodes [17]. By contrast, CD44high/ESAhigh cells had a phenotype similar to epithelial stem cells because they were highly proliferative, formed holoclone colonies in adherent cultures that were immortal and did not possess the ability to infiltrate lymph nodes [17]. Biddle et al. (2011) also demonstrated that the cells of one phenotype could regenerate cells of the other phenotype through reciprocal processes of EMT and MET and this phenotypic plasticity was maintained in vivo. It appears, therefore, that CSCs can adopt an EMT phenotype to migrate to secondary sites and then revert back to a proliferative non-EMT phenotype to establish tumour formation in the metastatic deposit [17]. These two distinct pathways of differentiation in CSCs, therefore, first facilitate the spread of the primary tumour and then, enable tumour formation at a distant site [17]. The same authors later showed that increased phenotypic plasticity (i.e., the ability to undergo EMT/MET) underlies increased CSC therapeutic resistance within both the epithelial and post-EMT sub-populations. The post-EMT CSCs that possess plasticity exhibit particularly enhanced therapeutic resistance and are defined by a CD44high/ECAMlow/CD24+ cell surface marker profile [53].

The findings of Biddle et al. (2011) are entirely consistent with the results of Chen et al. (2011) who showed that CSCs derived from spheroid forming cells had a decreased E-Cadherin expression and an increased expression of Snail, both of which are indicative of an EMT phenotype [30]. Further, Davis et al. (2010) showed that CD44+ cells had increased motility which is consistent with an EMT phenotype; these cells metastasised in vivo, whereas non-CSCs failed to metastasise [36]. Shigeishi et al. (2013) have also described the characteristics of ESA cell types; ESAlow CSCs had a fibroblast-like phenotype, showed an increased expression of Snail, Vimentin, and Axl, and demonstrated a low expression of E-cadherin; ESAhigh CSCs formed holoclones, expressed high levels of E-cadherin, and low levels of Snail and Vimentin which resulted in a faster growth rate in adherent culture conditions than CD44high/ESAlow cells [33]. In a similar way, Seino et al. (2016) described CD44/ALDH1 cells that displayed a spindle shape and showed a high expression of EMT-related genes, suggesting an EMT phenotype [27].

EMT-related genes may also play a role in the tumorigenic potential of HNCSCs. Chen et al. (2009) demonstrated that EMT-related genes were upregulated in both ALDH1+ and CD44+/CD24-/ALDH1+ cells; further, the knockdown of Snail significantly reduced sphere formation, tumorigenicity in vivo, and colony formation in vitro in both ALDH1+ and CD44+/CD24-/ALDH1+ cells, suggesting that Snail expression plays a key role in the regulation of self-renewal, tumorigenesis, and cancer stem properties in malignant HNSCC tumours [11]. Entirely consistent with these findings, Mohanta et al. (2017) showed that poorly differentiated HNSCCs had a significantly higher expression of the EMT markers Vimentin, Snail, and COL3A1, together with a significantly elevated EMT gene expression of N-cadherin, MMP2, and MMP9, which are able to initiate stromal degradation [38].

Taken together, the data indicate that CSCs in HNSCC are plastic and have the capacity to acquire an EMT phenotype that promotes tumour invasion and migration and also, the ability to revert to a more proliferative phenotype that facilitates tumour growth at the site of the metastatic deposit.

### 3.4. Role of Endothelial Cells and EGF in Promoting EMT and Maintaining Stemness of CSCs

Similar to normal stem cells, CSCs depend on their immediate microenvironment for their survival and function, a phenomenon termed a “stem cell niche”. In the main, this niche consists of cellular and non-cellular components that support the survival, and regulate the proliferation, of CSCs [6] (Appendix A).

The cellular components of the stem cell niche relate predominantly to endothelial cells. Krishnamurthy et al. (2010) and Zhang et al. (2014) showed that ALDH+ cells in HNSCC are located in close proximity to blood vessels [6,49] where endothelial cell-derived growth factor enhances the expression of Bmi-1, increases spheroid formation, and enhances tumour growth in vivo; selective ablation of blood vessels reduces the proportion of CSCs in vitro and in vivo, suggesting that endothelial cells are able to secrete factors that promote the proliferation, survival, and self-renewal of HNCSCs [28,49]. Further, Zhang et al. (2014) found that epithelial tumour cells that are approaching a blood vessel undergo EMT (low E-cadherin, high vimentin), whereas once the tumour cells have entered the blood vessel, they revert back to an epithelial phenotype which can be explained by the phenomenon of MET, also reported by Biddle et al. (2011). These findings underscore the plasticity of these tumour cells [17].

With respect to the non-cellular components of the stem cell niche, Xu et al. (2017) found that epidermal growth factor (EGF) was able to promote the acquisition of stemness factors and induce EMT because the stimulation of HNSCC cell lines with EGF resulted in a decreased expression of E-cadherin and a simultaneous increase in the expression of vimentin [48]. EGF stimulation also increased the proportion of CD44+ cells in a representative cell population and enhanced the expression of Bmi-1 and ALDH1, thereby generating stem-like cells. The presence of EGF has also been shown to enhance the expression of ALDH1, vimentin, and PDK1, together with increasing the incidence of cervical LN metastasis in xenograft models [48]. Interestingly, EGF-induced EMT and CSC-like cell properties depend on aerobic glycolysis because the inhibition of glycolysis abolishes EGF-induced EMT and the acquisition of CSC-like phenotypes in OSCCs both in vitro and in vivo, phenomena that are possibly related to the upregulation of PDF1 [48]. Thus, the activation of EGF/EGFR signalling pathways facilitates the EMT process and enriches ALDH+/CD44high CSC-like cells with increased invasiveness and metastasis both in vitro and in vivo.

The question remains as to the signalling pathways that EGF adopts to elicit these cellular changes. Zhang et al. (2014) showed that EGF induces Snail through the PI3K-Akt pathway; EGF and endothelial cell-secreted factors activate STAT3, Akt, and ERK, whereas the inhibition of PI3K/Akt signalling prevents EGF- and endothelial cell-induced Snail expression [49]. These changes induce EMT and lead to the acquisition of stemness in HNSCC cell lines as shown by: (1) the downregulation of epithelial markers (E-cadherin, desmoplakin), (2) the upregulation of mesenchymal markers (vimentin, N-cadherin), (3) the induction of cell motility, (4) the enhanced expression of ALDH and CD44, and (5) the growth of cells as non-adherent orospheres. By contrast, the silencing of EGF in endothelial cells slows tumour growth and decreases the proportion of stem-like cells in xenograft models [49]. Interestingly, endothelial cell-derived factors can enhance CSC survival by protecting against anoikis (a form of programmed cell death) [29,49], which also involves the PI3K-Akt signalling [29]. Therefore, the induction of cell survival may be responsible, at least in part, for the ability of epithelial tumour cells to leave their nests and migrate through connective tissue [49]. Thus, the activation of EGF/EGFR signalling pathways, together with signals from tumour-associated endothelial cells, facilitate both the EMT process and the enrichment of CSC-like cells with an increased invasiveness, motility, and metastatic potential in vitro and in vivo.

### 3.5. Pathways Related to Resistance to Treatment and Immune Escape

HNCSCs are renowned for their resistance to radio/chemotherapy. This resistance is thought to be related to the upregulation of drug resistant genes including MDR-1, MRP-1, ABCG2 (ATP-binding cassette (ABC) transporter), and genes associated with the drug efflux pump [7,8,10,11,39]. A number of studies have contributed to this conclusion and, in so doing, have implicated different molecular pathways in radio/chemo-resistance. Chikamatsu et al. (2012) found that CD44+ cells, compared to CD44− cells, had lower rates of apoptosis in response to various apoptosis-inducing stimuli, including TNF-α, anti-Fas, and TRAIL. These CD44+ cells also had an increased expression of anti-apoptotic genes such as BCL2, BCL2A1, BCL2L1, BNIP1, and NAIP, which are known to provide protection against apoptosis-inducing stimuli and radio/chemotherapy [41]. Bourguignon et al. (2012) also reported that the CD44/HA pathway was involved in the anti-apoptotic properties of HNCSCs [20]. IAP (inhibitors of apoptosis) proteins (cIAP-1, cIAP-2, and XIAP), for example, are up-regulated in CD44v3high/ALDH1high cells following HA treatment, thereby enhancing the anti-apoptotic effects and decreasing the ability of cisplatin to induce apoptosis and cell death. These findings are consistent with the work of Chen et al. (2009) who showed that the knockdown of Snail significantly reduced cisplatin resistance in both ALDH1+ and CD44+/CD24−/ALDH1+ cells.

Other molecular pathways have also been implicated in HNCSC radio/chemo-resistance. The inhibition of the Wnt pathway by the Wnt antagonist sFRP4 disrupts sphere formation in vitro, and the treatment of CSCs with sFRP4 plus cisplatin reduces the expression of ABC genes (ABCG2 and ABCC4) with a consequent reduction in cell viability and the induction of apoptosis by the activation of the caspase 3/7 enzyme [46]. Similarly, Lee et al. (2014) found that the overexpression of β-catenin enhanced resistance to cisplatin treatment [47]. Lim et al. (2014) showed that the knockdown of c-Met also resulted in a reduced expression of ABCG2, whilst Sun and Wang (2011) reported that c-Met+ cells were resistant to cisplatin treatment [21,22].

With respect to escape from immunotherapy, Lee et al. (2017) demonstrated that a major mechanism underlying the immunosuppressive nature of CD44+ cells in HNSCC was the greater constitutive and inducible expression of PD-L1 [16]. PD-L1 binds to an inhibitory receptor, the programmed death 1 receptor (PD-1), which is expressed on activated T cells; the binding of PD-L1 to the PD-1 receptor results in the inhibition of T cell receptor-mediated activation. Further, the binding of PD-L1 to CD80 prevents the activation of CD80 by CD28 and this, in turn, contributes to the immunosuppressive activity of PD-L1 [16]. Interestingly, Lee et al. (2017) showed that PD-L1 is preferentially expressed on CD44+ cells, which also have higher levels of ZEB1 and STAT3 phosphorylation, both of which are known to regulate PD-L1 expression [16].

CD44+ expression in HNCSC appears to play a unique role in the downregulation of antitumor immunity. CD44 inhibits the proliferation of T-cells activated by anti-CD3/anti-CD28, it enhances the induction and recruitment of T-regulatory cells which downregulate immune responses and it inhibits IFN-γ and IL-2 production in PHA-activated PBMCs which leads to the inhibition of TH1 responses [51]. CSCs also produce higher levels of a variety of cytokines including IL-8, GCSF, and TGF-β, which impact on tumour cell proliferation, angiogenesis, invasion, and metastatic dissemination [51]. Additionally, it has been suggested that CD44+ cells may be less immunogenic than CD44− cells because CD44+ cells express less HLA-A2 and HLA class II antigens compared to CD44− cells and PBMC, suggesting that CD44+ cells may not be recognised as readily by either cytotoxic T lymphocytes or natural killer cells [51]. Further, TAP2 is also expressed in CD44+ cells, indicating that there may be a defect in the antigen-presenting machinery of these cells [51].

Taken together, HNCSCs have the capacity to evade the immune system using a variety of methods that include a reduction in immunogenicity, an inhibition of immune activation by the inhibition of TH1 responses, and the induction of immune suppression by the recruitment of T-regulatory cells.

## 4. Limitations of Published Work

We are aware that there are limitations to the body of work that argues that CSCs are key factors in tumorigenesis. CSC heterogeneity, at the moment, is largely defined by the expression of molecular markers but rarely have differences in the expression of such markers been carried out in direct comparative studies on the same tumour samples. Instead, the expression of CSC markers has been reported in parallel studies by different research groups. Variation in the expression of such markers, therefore, does not necessarily reflect different cell functions.

The majority of studies rely solely on in vitro studies and where in vivo work has been undertaken to determine gene function, it is necessarily confined to mouse models. In vitro models of carcinogenesis are often unable to replicate the complex tumour environment in humans and the results of animal models are not always transferable. Thus, the experimental findings in these studies may not reflect the true nature of the biological mechanisms by which CSCs drive tumorigenesis, metastasis, and treatment resistance in HNSCC. In addition, some of the published articles use only a single cell line or primary tumour samples obtained from a small number of patients, thereby reducing the power of the experimental data. Finally, the majority of clinical studies involve patients with HNSCC which, clinically and pathologically, are a significantly diverse group of tumours.

In the main, the published work has focussed on CD44 as a marker of HNCSCs; numerous other markers have received relatively little attention [3]. A single molecular marker that exclusively identifies cells with CSC-like properties has not been found to date. The identification of a specific marker of CSCs, however, would be an important starting point to facilitate functional studies of different CSC subpopulations. Whilst CD44 has been used extensively as a ubiquitous marker of CSCs, its importance is now being questioned in HNSCC. CSCs are thought to represent a minor sub-population in most cancers and, indeed, CD44+ CSCs are believed to constitute some 10% of the tumour cell population [18]. Reports by other groups [54,55], however, do not corroborate the notion that CD44+ cells are a minor subpopulation in HNSCCs. In these studies, either the majority of tumour cells express CD44 or no differences in the number of CD44 cells are evident between normal mucosa and primary HNSCC. A study by Sharaf et al. [55], for example, showed that the CSC markers BMI-1 and BCL11B, discriminated between healthy and cancerous tissues, whereas ALDH1A1 and CD44 were expressed to a comparable extent in normal and malignant tissues. We believe that caution needs to be exercised in the interpretation of these data because more than half of the HNSCCs in a study conducted by Sharaf et al. (2021) were HPV+ and the sample size was extremely small [55].

The presence or absence of CD44 can be explained, albeit in part, by the concept of an inducible spatio-temporal expression of the marker and also, by an isoform switch. For example, Gomez et al. (2020) demonstrated a link between the microenvironment and CD44 expression in CSCs. Specifically, they showed that tumour-associated macrophages influenced CD44 expression and mediated stemness via the PI3K-4EBP1-SOX2 pathway; further, a CD44 isoform switch regulated EMT plasticity [56]. Notwithstanding these observations, it appears that the expression profiles of specific markers in CSCs may be more complex than first anticipated. In ovarian cancers, for example, there is an increased expression of the mesenchymal spliced variant CD44s (standard) and a concurrent decrease in the epithelial variant (CD44v); during TGFβ1-induced EMT, there is a predominance of CD44s [57].

## 5. Concluding Remarks

Tumours develop by a succession of changes in cellular genomes and epigenomes that confer growth advantages to certain cell populations. This, in turn, leads to the clonal expansion of those cells that express these genotypes and phenotypes until, eventually, a primary tumour develops. In the epithelium, some argue that these changes occur in non-stem cell transit amplifying cells, whilst others support the notion that the alterations occur in stem cells.

In the present paper, we put forward the argument that CSCs are key drivers of tumorigenesis, metastasis, and treatment resistance in HNSCC. Given that the functionality, universality, and clinical credibility of the CSC phenotype is based solely on the so-called markers of stemness, functional tests are increasingly being used to select CSC-enriched cell populations. Side Population (SP) cells, as defined by an exclusion of the Hoechst dye in flow cytometry experiments, have been identified in many solid tumours and cell lines and the SP phenotype is increasingly being used to isolate CSCs and/or to provide an enriched source of CSCs. Another functional technique that leads to the enrichment of CSCs is the multi-cellular tumour spheroid model and, in this case, it is particularly well suited to anticancer compound screening [58]. Interestingly, CSC-directed chemotherapy is also in use in clinical trials involving recurrent/resistant tumours. It is believed that the ability to accurately isolate cells with CSC-like properties may be fundamental to the development of personalised cancer interventions.

Despite major advances in our understanding of the role of CSC heterogeneity in carcinogenesis, to date this knowledge has not been translated into new diagnostic criteria and therapeutic modalities. We believe that it is entirely valid to suggest that targeting the pathways and proteins that are essential to the survival of CSCs and the maintenance of stemness may prove to be a promising therapeutic approach. Given the global burden of HNSCC and its poor prognosis, understanding the pathobiology of CSCs is vital to improve treatment outcomes and survival rates. Our current understanding of CSCs is incomplete, and much work still needs to be undertaken to understand the true nature of CSCs and the biological processes that drive their behaviour and function.

## Data Availability

Not applicable.

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
