# Peer review of "Heterogeneity of Cancer Stem Cells in Tumorigenesis, Metastasis, and Resistance to Antineoplastic Treatment of Head and Neck Tumours"

_cells, 2021, doi:10.3390/cells10113068_

Round 1

Reviewer 1 Report

In their review manuscript, Wu and Cirillo aim at reviewing the current knowledge on the heterogeneity of cancer stem cells in head and neck cancers. While the topic is interesting and timely, the authors should have provided more depth to the review. Certain aspects remain superficial and require a more detailed information and support by more recent publications.

Major points:

  1. Language requires revision. Example: In the Abstract the following sentences are grammatically wrong and substantially disturb reading and understanding of the manuscript:
    1. “Diverse sub-populations HNC stem cells (HNCSCs) have been previously identified CSC specific markers, the most common…”.
    2. “Interestingly, distinct HNSCSC subsets play a differential roles in the generation…”
  2. The authors define heterogeneity in CSC based on differences in marker expression. However, such differences are rarely the result of a direct comparative study on the very same tumor samples but are more likely coming from parallel reports by different research groups. Hence, such notion must be described and discussed with proper caution as differently chosen markers might not automatically entail different functions.
  3. In Chapter 2, the description of CSC, their markers and functionality remain very superficial, sometimes only referring to markers within tables without discussing the markers themselves or entirely without any reference to the actual markers. The level of information for the reader is low and lacks the depth expected in a review article on such specific topic.
    1. “HNCSCs identified via markers or SP assays have been shown to have enhanced metastatic potential in vivo.”. Which markers were used? How was the metastatic potential defined? Distant or loco-regional metastases?
    2. “Several studies have also reported an association between enhanced expression of stem cell markers in patient tumors with lymph node metastasis compared to primary tumors though the difference was not always significant.” Firstly, the sentence is ambiguous in its meaning (stem cells markers were probably assessed in primary tumors for patients with and without LN mets, but that is unclear here). Secondly, no markers are mentioned, and the information is too broad. Thirdly, the differences do not even appear to be significant.
    3. “…and demonstrated a level of drug resistance to several chemotherapeutic agents…”. This is very vague and not informative.
    4. “The presence of CSC signatures was also reported to be significantly correlated with incomplete response …”. Which signatures? As such, this remains very superficial.
  4. In Chapter 3, despite a description of singular stem cell markers the overall novelty and content of the review remains comparably low. In this chapter, the choice in describing certain markers, pathways, and differentiation programs is not explained and lacks cohesion regarding the entire manuscript.
  5. The description of work by Biddle et al. and others is interesting in the context of EMT/CSCs and could give a blueprint for a major overhaul of the manuscript´s structure with the aim to become more specific and precise.
  6. EGF-dependent EMT in head and neck cancers was more recently covered by numerous groups including Sankpal et al. (PMID: 28192403), Pan et al. (PMID: 30261040), Gao et al. (PMID: 30132555).
  7. Generally, the authors focus considerably on CD44 expression as a marker for HNSCC CSCs. Firstly, numerous other markers have been reported and get little attention in this review. Secondly, as the authors mention CSCs are believed to represent a rather minor sub-population in most cancers. As claimed by Prince et al., CD44+ CSC represent a minor population (typically less than 10%). However, representative staining of CD44 in Figure 5B of Prince et al. and reports by Mack et al. (PMID: 18852874), Sharaf et al. (PMID: 34287293) and others, do not corroborate the notion of a minor sub-population of CD44+ cells within HNSCC. In fact, the majority of tumor cells expressed CD44 and/or no differences could be observed between normal mucosa and primary tumors of the head and neck. Hence, such discrepancy should be addressed including more recent publications such as Gomez et al. (PMID: 32816856) on the molecular link of CD44 and HNSCC CSC.

Minor points:

  1. HNSCC (head and neck squamous cell carcinomas) is a more widely used term as it encompasses roughly >95% of the malignancies in this localization. Unless the authors specifically aim at implementing adenomatous carcinoma of the head and neck too, I suggest using HNSCC. Alternatively, the authors should address this difference in the manuscript more precisely.
  2. Page 2, line 72: “By and large,…”. Sentence is incorrect.
  3. Page 8, line 262: “This cell population thus appears to have an EMT phenotype and were able to infiltrate lymph nodes”. Sentence is incorrect.
  4. Page 8, line 287-292: Sentence is incorrect.

Author Response

please find attached our response to the criticisms raised.

Reviewer 2 Report

The paper by Wu and Cirillo describes the importance of stem-like cancer cells in head and neck tumors with regard to tumorigenesis, metastasis and resistance. The paper is clearly written, well organized and incorporates latest findings in this field. The tables which accompany the text provide succinct lists of findings. Especially the second part of the paper (starting from par. 3) is interesting.

Line 27 - I would change/discard the phrase "in the early 20th century" because the stem cell hypothesis was proposed much later and became dominant in the 21st century.

The paper could be improved if the Authors discussed the significance of surface or intracellular stem markers in greater detail with regard to prevalence, conditions of detection or dynamic changes. A nice paper investigating stemness markers in HN cells grown in 3D is: Goričan L, Gole B, Potočnik U. Head and Neck Cancer Stem Cell-Enriched Spheroid Model for Anticancer Compound Screening. Cells. 2020;9(7):1707.

Also, it would be interesting to add information regarding different variants of CD44 (like the mentioned CD44v3) in HNCSC.

Moreover, do the Authors have any suggestions or recommendations regarding the investigation of stem cell markers? What should be performed or what should be avoided in such analyses?

Line 173 - what do the Authors mean by 'product' in "cMET is a product of transmembrane tyrosine kinase..."?

Paragraph 3.2. - the Authors refer to canonical Wnt pathway (line 211 - please add 'canonical'). A recent review [Paluszczak J. The Significance of the Dysregulation of Canonical Wnt Signaling in Head and Neck Squamous Cell Carcinomas. Cells. 2020;9(3):723] nicely presented all important findings regarding this pathway in HNSCC and it could be used as a reference.

Please, pay attention to nomenclature - use 'Wnt' and not 'wnt'. Please, also check the names of genes so that their symbols are properly written according to directions for writing names of human genes.

Author Response

(The authors gave the same response as above.)

Round 2

Reviewer 1 Report

The authors have addressed some of the points of criticism and have chosen to not included others. The authors also mention potential differences in writing styles that might results in slightly diverging views by the authors and reviewers. This should obviously be respected.
I consider the manuscript valuable and worth of publication. 
Thank you very much for the changes made during the revision.